# Fasciculation potentials are related to the prognosis of amyotrophic lateral sclerosis

**Keiko Ohnari[1]\*, Kosuke Mafune[2], Hiroaki Adachi[1]\***

1 Department of Neurology, School of Medicine, University of Occupational and Environmental Health, Kitakyushu, Fukuoka, Japan, 2 Department of Mental Health, Institute of Industrial Ecological Sciences, School of Medicine, University of Occupational and Environmental Health, Kitakyushu, Fukuoka, Japan

\* keiko-o@med.uoeh-u.ac.jp (KO); hiadachi@med.uoeh-u.ac.jp (HA)

## Abstract

Some prognostic biomarkers of amyotrophic lateral sclerosis (ALS) have been described; however, they are inadequate for satisfactorily predicting individual patient outcomes. Fasciculation potentials (FPs) on electromyography (EMG) are useful for the early diagnosis of ALS, and complex FPs are associated with shorter survival in ALS. In this study, we investigated the relationship between the proportion of muscles with FPs, biochemical markers, and the prognosis of ALS. 89 Patients with ALS were retrospectively classified into three groups based on the interval from onset to death or tracheostomy (less than 1 year: fast progression; from 1 year to less than 3 years: average progression; 3 years or more: slow progression). We performed statistical analysis of the electrophysiological findings, including the percentage of examined muscles with FPs, and biochemical markers evaluated on admission. Patients with fast ALS progression had a higher percentage of muscles with FPs (93.1% vs. 37.9%, P<0.001) and lower uric acid (UA) levels (male: 4.19 mg/dl vs 5.55 mg/dl, P<0.001; female: 3.71 mg/dl vs 5.41 mg/dl, P<0.001) than patients with slow progression. Survival curves demonstrated a relationship between these factors and the survival time in patients with ALS. Furthermore, UA levels were correlated with the percentage of muscles with FPs. Our electrophysiological findings suggest that ALS presents with multisystem neurological manifestations, and these manifestations differed among the groups classified by disease progression. The percentage of muscles with FPs on EMG and serum UA levels were especially associated with the prognosis of ALS.

## Introduction

Amyotrophic lateral sclerosis (ALS) is a progressive neurodegenerative disease characterized by upper and lower neuronal manifestations. However, its pathogenesis remains unclear. The average survival time is 2–4 years, but disease progression varies among patients [1]. The cumulative time-dependent survival rates at 1, 5, and 10 years after diagnosis have been reported to be 76.2%, 23.4% and 11.8%, respectively [2]. Many clinical trials for ALS have been conducted and are ongoing; hence, the features and prognostic factors of ALS need to be adequately clarified to assess the effectiveness of treatment modalities. Based on data collected from patients with ALS across Europe, bulbar onset, autonomic dysfunction, diagnosis of

**Data Availability Statement:** All relevant data are within the manuscript and its Supporting information files.

**Funding:** The author(s) received no specific funding for this work.

**Competing interests:** The authors have declared that no competing interests exist.

definite ALS according to the revised El Escorial criteria, diagnostic delays, reduced forced vital capacity, progression rate, and presence of frontotemporal dementia have been reported as poor prognostic factors [3, 4].

The prognosis of ALS has been studied in terms of not only clinical symptoms but also electrophysiological findings and biochemical markers [5–7]. The correlation between the prognosis of ALS and electrophysiological findings in the central and peripheral nervous systems has been reported [8]. In particular, needle electromyography (EMG) findings, including motor unit potential morphology and the localization and frequency of fasciculation potentials (FPs), have been reported to be important for diagnosing ALS [5]. The methods for detecting FPs include conventional EMG and muscle ultrasound (MUS) [9]. MUS is not an invasive examination and can facilitate real-time detection of FPs from a large body surface area. FPs analyzed using MUS have been reported to show prognostic significance for ALS [10, 11]. However, MUS does not allow analysis of the morphology or firing characteristics of the motor units involved [12]. Moreover, MUS findings are not included in the Awaji and Gold Coast criteria [13–15].

The prognostic relationship between neurofilaments and biochemical markers such as creatinine, uric acid (UA), and lipid levels has also been reported previously [16–18]. and neurofilaments have been reported to be diagnostic biomarkers [18]. However, compared with neurofilament analyses, the analyses of creatinine and UA are cheaper and can be used as prognostic biomarkers of ALS.

Given the aforementioned limitations of MUS and prospects of creatinine and UA as prognostic markers, this study aimed to clarify the prognosis of ALS using needle EMG, and also to study the association between electrophysiological findings and biochemical markers for the prognosis of ALS.

## Methods

### Patients

We studied 89 patients with ALS who were treated at the Hospital of the University of Occupational and Environmental Health, Japan, between May 11, 2000 and December 31, 2023. All patients were diagnosed with sporadic, definite ALS according to the revised El Escorial and Awaji criteria [13, 14]. and presented with progressive disease. The median survival duration of the patients with ALS was 3 years, and the 1-year survival rate was 76.2% [2]. Based on the interval between symptom onset and death or tracheostomy, the patients were retrospectively classified into three groups (less than 1 year: fast progression; from 1 year to less than 3 years: average progression; and 3 years or more: slow progression). We also investigated the period from onset to first evaluation and the symptoms at onset. Biochemical tests, including the measurement of UA, albumin, total cholesterol, creatine kinase, triglyceride, and creatinine levels, and neutrophil count were performed at diagnosis. The study design was approved by the Ethics Committee of Medical Research at the University of Occupational and Environmental Health, Japan (UOEHCRB20-021). The study conforms with World Medical Association Declaration of Helsinki. Informed consent was not obtained from study participants as the study did not involve a prospective evaluation. The Ethics Committee of the University of Occupational and Environmental Health granted a permission to use the retrospective data in the study without individual informed consent. The authors had access to information that could identify individual participants during and after data collection. The data for our study was accessed on January 1, 2024.

## Electrophysiological studies

All patients underwent nerve conduction studies (NCS) and needle EMG (Neuropack; Nihon Kouden, Tokyo, Japan). Motor conduction studies were performed in the median, ulnar, tibial, and peroneal nerves, for which recordings were obtained from the abductor pollicis, abductor digiti minimi, extensor digitorum brevis, and abductor hallucis, respectively. Sensory conduction studies were performed in the median, ulnar, and sural nerves. Recordings were obtained from the proximal interphalangeal joint of the index finger, proximal interphalangeal joint of the little finger, and area just behind the lateral malleolus. We ruled out polyradiculoneuropathy using NCS. Needle EMG was performed on the muscles of the upper extremities (UEs) and lower extremities (LEs), and the paraspinal and trapezius muscles. As tongue relaxation was difficult and fibrillation potentials, positive sharp waves (Fib/PSWs), and FPs were more frequently observed in the trapezius muscle than in the tongue in patients with ALS [19]. we performed EMG in the trapezius muscle. The most frequently examined muscles of the extremities in different segments were the deltoid, biceps brachii, triceps brachii, first dorsalis interossei, rectus femoris, anterior tibialis, and gastrocnemius. The number of muscles examined was determined based on the diagnostic implications. We examined the occurrence of Fib/PSWs and FPs at rest. FPs were defined as spontaneous and random motor unit potentials showing a highly complex morphology with reproducibility over observation for up to 90 seconds [20]. Therefore, we analyzed the proportion of the examined muscles that had Fib/PSWs and FPs.

## Statistical analysis

Age, interval from onset to first evaluation, biochemical test results, percentage of muscles with Fib/PSWs and FPs, and NCS results were compared among the groups using analysis of variance. The variables in the NCS include motor nerve conduction velocities, distal latencies, compound muscle action potentials, sensory nerve conduction velocities, and sensory nerve conduction potentials. Sex and symptoms at onset were analyzed using Fisher's exact test. The relationships between these variables and percentage of FPs were analyzed using correlation coefficient and a multiple linear regression model. Survival analysis was performed using Kaplan–Meier method with a log-rank test, the variables being the percentage of FPs and biochemical markers. Furthermore, Cox proportional hazards regression models were used to adjust for other factors affecting the percentage of FPs and UA levels. The covariates included in the model were age, sex, and symptoms at disease onset. The percentage of FPs was classified into three groups: less than 50%, 50–99%, and 100%. UA level was classified into three groups for both sexes: for males, <4.7, 4.7–5.4, and >5.4 mg/dl; for females, <3.7, 3.7–4.4, >4.4 mg/dL. The upper limit of the average survival time was defined as 48 months. Linear regression was used to examine the correlation between the percentage of FPs and UA levels. P-value <0.05 indicated statistically significant difference among the three groups.

## Results

Based on the interval from ALS onset to death or tracheostomy, the total number of patients with ALS was 89. There were 29, 28, and 32 patients with fast, average, and slow progression, respectively (Table 1). The three groups showed no significant differences in sex distribution or rate of onset of extremity weakness. The age at onset did not differ between patients with fast progression and those in the other two groups; however, slow progression was more frequently observed among young patients. The percentage of patients with bulbar onset in the fast progression group was higher than that in the average progression group; however, there was no significant difference from the slow progression group. The time from symptom onset

**Table 1. Clinical findings in patients with ALS classified according to clinical course.**

| | Fast progression | Average progression | Slow progression | P-value | P-value, Fast progression vs Average progression | P-value, Fast progression vs Slow progression | P-value, Average progression vs Slow progression |
|---|---|---|---|---|---|---|---|
| Number of patients | 29 | 28 | 32 | | | | |
| Sex (men/women) | 19/10 | 15/13 | 15/17 | 0.337 | | | |
| Age at onset (years) | 67.8 ± 7.9 | 69.2 ± 9.0 | 62.4 ± 12.5 | **0.024*** | 0.855 | 0.099 | 0.028 |
| Time from symptom onset to the first evaluation (months) | 5.0 ± 2.3 | 11.3 ± 7.3 | 19.5 ± 10.9 | **<0.001*** | **0.009*** | **<0.001*** | **<0.001*** |
| Bulbar onset | 15 | 5 | 12 | **0.035*** | **0.049*** | 0.390 | 0.243 |
| UE onset | 10 | 16 | 10 | 0.091 | | | |
| LE onset | 7 | 8 | 10 | 0.825 | | | |

* statistically significant

ALS, amyotrophic lateral sclerosis; UE, upper extremity; LE, lower extremity

to the first evaluation in patients with fast progression was significantly shorter than in the other two groups (5.0±2.3, 11.3±7.3, and 19.5±10.9, respectively, P <0.001) (Table 1).

The EMG findings at rest are summarized in Table 2. The percentages of muscles with FPs in the fast, average, and slow progression groups were 93.1%, 62.9%, and 37.9%, respectively. Shorter disease period was significantly associated with more frequent FPs (P <0.001) (Table 2). Moreover, the survival curves obtained using the Kaplan–Meier method and Cox proportional hazards regression models demonstrated a relationship between the percentage of muscles with FPs and survival time in patients with ALS (P<0.000) (Fig 1A and 1B). In contrast, Fib/PSWs did not differ significantly among the three groups (S1 Fig). During the NCS, the motor conduction velocity (MCV) of the ulnar and tibial nerves in patients with fast progression was significantly lower than that in the other two groups (Table 3). The distal latencies of the tibial nerve in patients with fast progression were longer than those in the other two groups. Furthermore, the sensory conduction velocities (SCV) of the ulnar and sural nerves in patients with fast progression were significantly lower than those in the other two groups. The other components of the NCS showed no significant differences, and we found milder disturbances in patients with slow progression compared to the other groups (Table 3).

Table 4 presents a comparison of the laboratory data among the three groups. The UA levels in both male and female patients with fast progression were lower than those in the average and slow progression groups (male: 4.19, 4.89, and 5.55, P = 0.000; female: 3.71, 4.00, and 5.41,

**Table 2. EMG findings in patients with ALS classified according to clinical course.**

| | Fast progression | Average progression | Slow progression | P-value | P-value, Fast progression vs Average progression | P-value, Fast progression vs Slow progression | P-value, Average progression vs Slow progression |
|---|---|---|---|---|---|---|---|
| Muscles with FPs (%) | 93.1 ± 11.6 | 62.9 ± 26.4 | 37.9 ± 25.9 | **<0.001*** | **<0.001*** | **<0.001*** | **<0.001*** |
| Muscles with Fib/PSWs (%) | 34.1 ± 31.4 | 51.9 ± 27.6 | 42.9 ± 33.2 | 0.101 | | | |

* statistically significant

ALS, amyotrophic lateral sclerosis; EMG, electromyography; FPs: fasciculation potential; Fib/PSWs: fibrillation potential and positive sharp waves

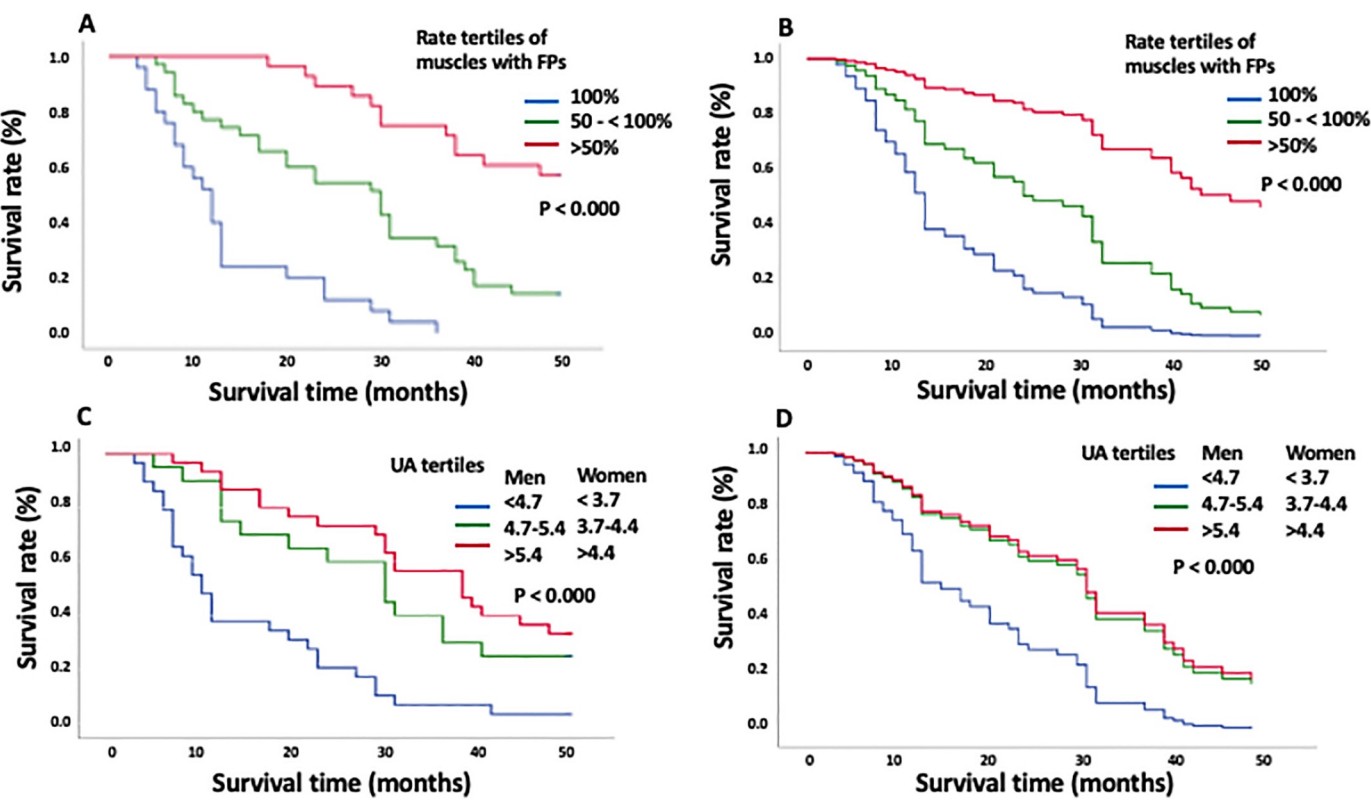

**Fig 1.** Comparison of the survival curves after disease onset, stratified by the percentage of muscles with fasciculation potentials (FPs) in patients with amyotrophic lateral sclerosis by using the Kaplan-Meier method (A) and Cox proportional hazard regression models (B). The survival curves obtained using Cox proportional hazard regression model showed a relationship between the percentage of muscles with FPs and survival time (B). Comparison of the survival curves after disease onset stratified by uric acid levels using Kaplan-Meier method (C) and Cox proportional hazard regression models (D). The survival curves of the Cox proportional hazard regression model showed significant differences between patients with low uric acid levels and the other groups (D).

P<0.001). Furthermore, the survival curve obtained using the Kaplan-Meier method demonstrated a relationship between UA levels and survival time in patients with ALS (P<0.001) (Fig 1C). The survival curves of the Cox proportional hazards regression model showed significant differences between patients with low UA levels and patients in the other two groups (P<0.000) (Fig 1D). The UA levels were significantly associated with FP (Table 5, Fig 2); this tendency was also observed in multiple regression analysis (Table 5). Laboratory parameters other than UA levels did not differ among the three groups (S2–S7 Figs). The triglyceride and creatine kinase levels of patients with fast progression were lower than those of patients with average progression, but not significantly lower than those of patients with slow progression.

## Discussion

This study aimed to clarify the prognosis of ALS by using needle EMG and assessing the association between electrophysiological findings and biochemical markers. Our electrophysiological study of the peripheral and central nervous systems demonstrated significantly different findings among the groups classified according to the prognosis of ALS. We found that patients with rapidly progressing ALS had higher percentage of muscles with FPs and lower UA levels. UA level as a biochemical marker is related to survival in patients with ALS. In particular, the percentage of muscles with FPs on EMG was associated with ALS prognosis, and

**Table 3. Nerve conduction studies of patients with ALS classified according to the clinical course.**

| | Fast progression | Average progression | Slow progression | P-value | P-value, Fast progression vs Average progression | P-value, Fast progression vs Slow progression | P-value, Average progression vs Slow progression |
|---|---|---|---|---|---|---|---|
| Median | | | | | | | |
| MCV (m/s) | 51.8±5.9 | 49.9±5.7 | 53.3±4.4 | 0.075 | | | |
| DL (ms) | 4.4±0.9 | 4.7±1.3 | 4.3±0.7 | 0.202 | | | |
| CMAP (mV) | 4.4±3.0 | 3.1±3.3 | 5.1±3.7 | 0.072 | | | |
| SCV (m/s) | 48.8±5.1 | 50.0±8.1 | 53.1±6.2 | 0.087 | | | |
| SNAP (μV) | 20.4±8.9 | 20.3±12.0 | 25.5±10.3 | 0.144 | | | |
| Ulnar | | | | | | | |
| MCV (m/s) | 49.9±6.5 | 50.8±6.9 | 57.2±5.5 | **<0.001**\* | 0.879 | **0.001**\* | **0.001**\* |
| DL (ms) | 3.6±0.5 | 3.6±1.0 | 3.2±0.3 | 0.056 | | | |
| CMAP (mV) | 5.6±2.9 | 4.2±2.9 | 5.4±2.9 | 0.174 | | | |
| SCV (m/s) | 50.1±4.2 | 52.3±5.8 | 55.3±6.9 | **0.012**\* | 0.420 | **0.010**\* | 0.144 |
| SNAP (μV) | 17.6±9.0 | 17.2±9.3 | 21.5±10.6 | 0.212 | | | |
| Tibial | | | | | | | |
| MCV (m/s) | 42.4±3.7 | 44.3±5.3 | 46.6±4.3 | **0.007**\* | 0.303 | **0.005**\* | 0.162 |
| DL (ms) | 4.9±0.9 | 4.2±0.9 | 4.0±0.6 | **<0.001**\* | **0.010**\* | **<0.001**\* | 0.457 |
| CMAP (mV) | 7.0±4.5 | 9.2±5.1 | 8.0±5.4 | 0.296 | | | |
| Peroneal | | | | | | | |
| MCV (m/s) | 41.9±4.5 | 41.6±4.6 | 44.5±5.4 | 0.082 | | | |
| DL (ms) | 5.1±1.2 | 5.5±1.2 | 5.0±1.0 | 0.327 | | | |
| CMAP (mV) | 2.0±2.4 | 1.7±1.9 | 1.8±1.9 | 0.891 | | | |
| Sural | | | | | | | |
| SCV (m/s) | 45.5±4.6 | 47.4±5.8 | 50.8±4.3 | **0.001**\* | 0.359 | **0.001**\* | **0.037**\* |
| SNAP (μV) | 12.4±6.5 | 12.5±6.6 | 13.3±6.9 | 0.861 | | | |

\* statistically significant

MCV, motor nerve conduction velocity; DL, distal latency; CMAP, compound muscle action potential; SCV, sensory nerve conduction velocity; SNAP, sensory nerve conduction velocity

the percentage of muscles with FPs and serum UA level showed a significant negative relationship. We also found that the MCV of the ulnar and tibial nerves and the SCV of the ulnar and sural nerves in patients with fast progression were significantly lower than those in the other two groups. Our present data confirmed that the percentage of muscles with FPs on EMG and serum UA levels are important biomarkers for predicting the prognosis of ALS.

This study demonstrated that higher percentage of muscles with FPs is associated with shorter disease duration in patients with ALS. The source generator of FPs is the motor neuron or axon prior to its terminal branches [21]. FPs are associated with numerous disease processes affecting the lower motor neurons, such as ALS [21]. FPs detected with needle EMG were the

**Table 4. Laboratory data of patients with ALS classified according to clinical course.**

| | Fast progression | Average progression | Slow progression | P-value | P-value, Fast progression vs Average progression | P-value, Fast progression vs Slow progression | P-value, Average progression vs Slow progression |
|---|---|---|---|---|---|---|---|
| Albumin (g/dl) | 4.04 ± 0.37 | 4.12 ± 0.35 | 4.07 ± 0.36 | 0.702 | | | |
| T-cholesterol (mg/dl) | 195.57 ± 40.44 | 204.32 ± 30.21 | 193.86 ± 36.4 | 0.537 | | | |
| Males | 188.56 ± 38.09 | 194.79 ± 28.15 | 188.36 ± 31.87 | 0.847 | | | |
| Females | 208.20 ± 43.49 | 218.00 ± 30.63 | 199.69 ± 39.57 | 0.503 | | | |
| CK (U/L) | 148.96 ± 180.06 | 227.43 ± 238.69 | 203.93 ± 154.79 | 0.319 | | | |
| Males | 189.17 ± 210.16 | 317.07 ± 283.82 | 282.36 ± 173.41 | 0.254 | | | |
| Females | 68.56 ± 24.51 | 124.00 ± 112.14 | 135.31 ± 97.47 | 0.219 | | | |
| TG (mg/dl) | 87.81 ± 30.65 | 120.15 ± 58.22 | 106.77 ± 49.60 | **0.049**\* | **0.040**\* | 0.276 | 0.591 |
| Uric acid (mg/dl) | 4.05 ± 0.90 | 4.89 ± 1.38 | 5.55 ± 1.17 | **<0.001**\* | **0.026**\* | **<0.001**\* | 0.114 |
| Males | 4.19 ± 0.77 | 5.60 ± 1.11 | 5.71 ± 0.81 | **<0.001**\* | **<0.001**\* | **<0.001**\* | 0.956 |
| Females | 3.71 ± 1.14 | 4.00 ± 1.17 | 5.41 ± 1.42 | **0.006**\* | 0.875 | **0.014**\* | **0.022**\* |
| Creatinine (mg/dl) | 0.59 ± 0.21 | 0.61 ± 0.17 | 0.56 ± 0.19 | 0.678 | | | |
| Males | 0.63 ± 0.19 | 0.67 ± 0.17 | 0.61 ± 0.23 | 0.719 | | | |
| Females | 0.50 ± 0.24 | 0.53 ± 0.14 | 0.53 ± 0.16 | 0.921 | | | |
| Neutrophils (/μl) | 3526.04 ± 1508.33 | 3719.32 ± 1096.84 | 3118.79 ± 1330.28 | 0.248 | | | |

\* statistically significant

ALS, amyotrophic lateral sclerosis; T-cholesterol, total cholesterol; CK, creatine kinase; TG, triglyceride

**Table 5. Factors related to the frequency of fasciculation potential.**

| | Univariate analysis | | | | Multivariate analysis | | | |
|---|---|---|---|---|---|---|---|---|
| | B[§] | 95% CI | β[⁋] | P value | B[§] | 95% CI | β[⁋] | P value |
| UA[†] | -7.465 | 59.072–71.933 | -0.321 | 0.001\* | -6.608 | -13.766–0.551 | -0.279 | 0.070[†] |
| Sex | | | | | -14.020 | -26.146 –-1.894 | -0.229 | 0.024\* |
| Age at onset | | | | | 0.507 | -0.069–1.084 | 0.168 | 0.083[†] |
| Bulbar onset | | | | | 14.384 | -7.936–36.704 | 0.225 | 0.203 |
| UE onset | | | | | 15.772 | -5.092–36.637 | 0.253 | 0.136 |
| LE onset | | | | | -1.565 | -22.517–19.386 | -0.024 | 0.882 |
| Time from symptom onset to the first evaluation | | | | | -0.334 | -1.893 –-0.449 | -0.334 | 0.002\* |
| Interaction | | | | | | | | |
| UA × Sex | | | | | 0.049 | -5.857–13.734 | 0.049 | 0.730 |

[§] partial regression coefficient

[⁋] standardized partial regression coefficient

\* statistically significant <0.05 [†] statistically significant <0.1

[†] mean uric acid level

UA, uric acid level; UE, upper extremity; LE, lower extremity; CI: confidence interval

Sex is a dummy variable labeled 0 = men and 1 = women.

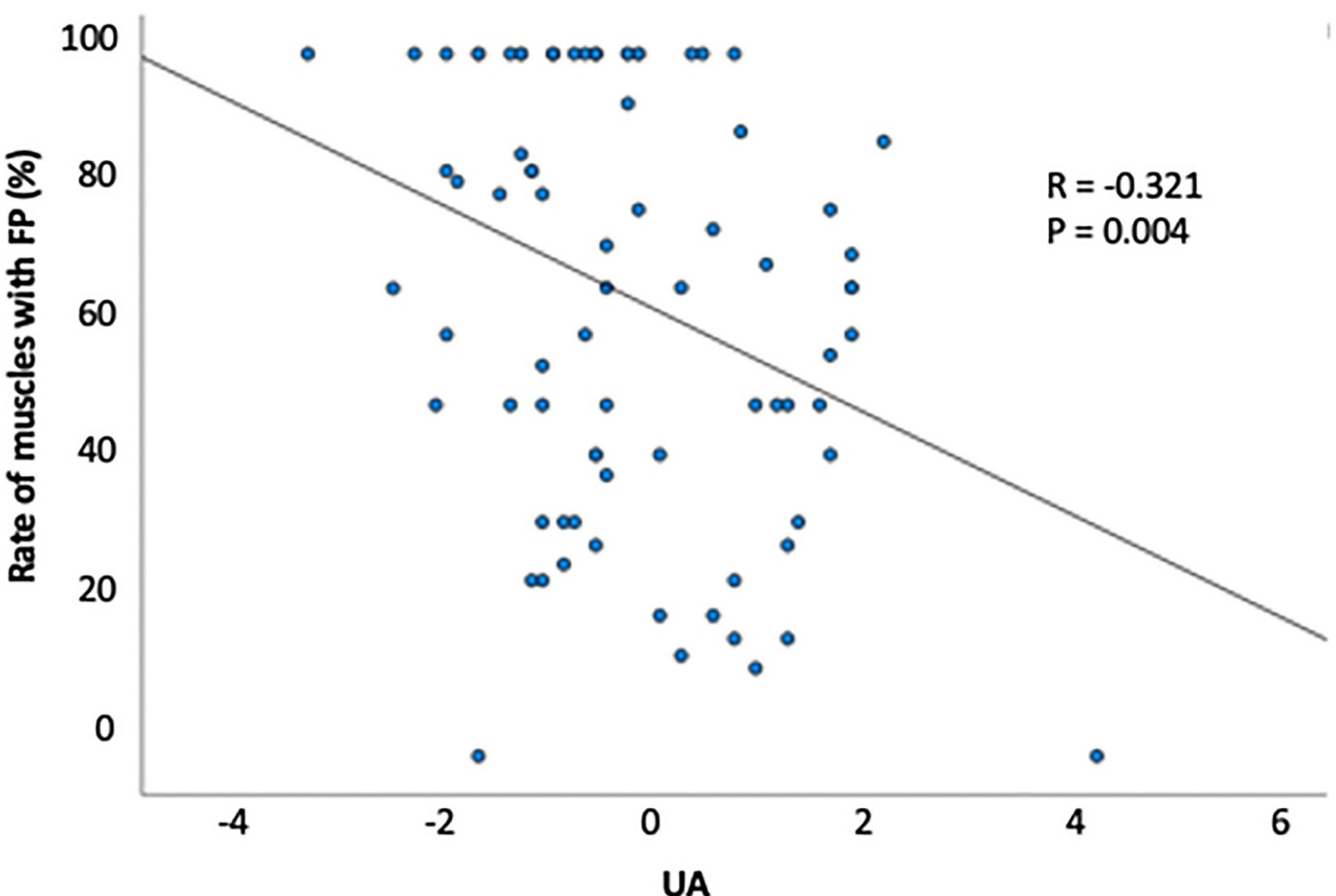

**Fig 2. Correlation between the percentage of muscles with fasciculation potentials and uric acid level (grand mean centered uric acid level).** The percentage of muscles with fasciculation potentials was negatively correlated with uric acid level (R = -0.321, P = 0.004).

core findings for diagnosing ALS since they reflect ongoing denervation in the affected muscles. Therefore, FPs and Fib/PSWs are equally important [14]. The rate and distribution of FPs have been reported to be specific findings in patients with ALS, in contrast to the findings of previous studies on polio, spinal and bulbar muscular atrophy, cervical spondylotic radiculopathy or myelopathy, lumbosacral radiculopathy, and multifocal motor neuropathy [22, 23]. In a previous study, the EMG findings in the trapezius muscle in cervical spondylosis showed neither Fib/PSWs nor FPs [19]. Moreover, the mean amplitude and duration of FPs increased with disease progression [24]. In contrast, the appearance of FPs is an extremely early marker of ALS, and FPs have been reported to occur in ALS without weakness and muscle atrophy [25].

The use of MUS for noninvasive diagnosis of ALS has been developed recently, and MUS and EMG have been reported to have nearly the same detection rates of FPs [10]. Several studies on MUS have shown that FPs are associated with the process and prognosis of ALS [6, 11, 12]. The rate of disease progression has been correlated with the number of fasciculations detected using MUS [11]. Furthermore, higher frequency of FPs in the biceps brachii and brachialis muscles with FPs on MUS was reported to be associated with shorter disease duration and faster decline in the ALS Functional Rating Scale-Revised (ALSFRS-R) score [6]. The features of FPs obtained using EMG in the ALS criteria have been compared to those obtained

using MUS [5, 12, 26, 27]. EMG is more sensitive than MUS in detecting fibrillation [26]. Moreover, MUS does not allow the analysis of morphology or firing characteristics of the motor units involved [12]. The morphology of FPs in EMG has been reported to be important for determining the diagnosis and stage of ALS [5, 12] and the presence of complex FPs was associated with shorter survival [5]. Small FPs may be undetectable with MUS because of the significantly lower mean amplitude of FPs in patients with EMG-detected FPs alone than in those with both FPs and MUS fasciculations (0.39 ± 0.25 mV and 1.22 ± 0.92 mV, respectively: $P < 0.0001$) [27]. In this study, using EMG, we found a higher percentage of muscles with FPs in patients showing fast progression than in the other two groups. In accordance with previous reports on MUS, these results indicate that FPs are related to the prognosis of patients with ALS [6, 11, 12].

In addition to needle EMG, electrophysiological studies, including NCS have also been performed for ALS diagnosis [14]. The results of NCS have been reported to be related to the prognosis of ALS [28]. Compound muscle action potential (CMAP) and sensory nerve action potential (SNAP) amplitudes of the median nerve were considered independent prognostic factors of ALS [28]. In our study, we did not find significant differences in the CMAP and SNAP amplitudes of the median nerves among the three groups, but the MCV, CMAP, SCV, and SNAP amplitudes of the median nerve in patients with fast progression tended to be lower than the corresponding values in patients with slow progression. The MCV of the ulnar and tibial nerves and the SCV of nerves other than the median nerve in patients with fast progression were shorter than those in the other two groups. These differences are related to poor prognosis in patients with ALS. Sensory abnormalities have also been reported to be common in NCS performed in patients with ALS [29, 30].

Although patients with ALS usually do not present with sensory disturbances, they may show subclinical dysfunction in the sensory nervous system [8]. The peak-to-peak amplitude between N20 and P25 in the median SEP has been associated with shorter survival in ALS [31]. In addition to needle EMG, NCS showed significantly different results between patients with ALS with fast progression and patients in the other two groups. These findings indicate that ALS is a neurodegenerative disease that affects not only the upper and lower neurons but also the multisystem of neurons, as reported previously [8]. The pathological findings obtained in the sural nerve biopsy of patients with ALS have been reported previously [32]. Pathological sensory disturbance detected through skin biopsies has also been reported for ALS, with observations including, a significant loss of intraepidermal nerve fiber (IFNF) and Meissner corpuscle density in ALS compared with healthy controls, and increasing IENF density over time were associated with a poorer prognosis [33]. Furthermore, the 43-kDa TAR DNA-binding protein identified to be related to ALS pathogenesis is found widely in the nervous system, including the upper and lower neurons [34]. Thus, based on its neuropathology, ALS can be considered a form of multisystem neurodegeneration rather than a pure motor neuron disease.

UA has been identified as a natural antioxidant and free radical scavenger in the processes underlying oxidative stress [35]. UA is also reportedly related to some neurodegenerative diseases with underlying pathogenesis involving oxidative stress, such as Parkinson's disease [36]. The pathogenesis of ALS is related to not only electrophysiological findings but also biomarkers such as UA [7, 37]. Oxidative stress has been proposed to play a role in the pathogenesis of ALS [38]. In one study, administration of edaravone, a potent radical and peroxynitrite scavenger, significantly delayed disease progression compared to what was observed in untreated patients [39]. Furthermore, the plasma levels of UA have been reported to increase in treated ALS patients [39]. Treatment with inosine was shown in another study to elevate the serum urate levels and slow down the progression of ALS, which was characterized using ALSFRS-R total scores [40]. The serum UA levels of patients with ALS were

previously reported to be lower than those in healthy individuals [37]. and shown to be associated with prolonged survival in ALS [41, 42]. Similar to previous reports, we found that serum UA levels were associated with prolonged survival in ALS. Moreover, univariate analysis showed a significant negative correlation between the percentage of muscles with FPs and UA levels, whereas multivariate analysis showed a significant relationship. No reports on the relationship between FPs and serum UA levels s in ALS have been published to date. However, serum UA levels and FPs were negatively correlated in this study. This suggests that serum UA levels s may be involved in the mechanism underlying electrophysiological changes that lead to FPs in skeletal muscle and may modify the pathology of ALS. Although the association between prognosis and UA level was only reported in male patients previously [43]. UA level was subsequently reported to be a prognostic factor without significant differences between the sexes [44]. Albumin and creatinine levels have also been reported to be prognostic biomarkers of ALS. In contrast, lipid levels and neutrophil counts have recently been reported to be unrelated to disease progression [45, 46]. Our findings did not show any association between lipid levels, neutrophil counts, and ALS prognosis. Thus, UA level can be presumed to be an important prognostic factor for ALS. Sporadic ALS is an intractable disease, and its mechanism of onset is still unknown. The discovery of several factors involved in the pathogenesis or modification of the disease progression of ALS may lead to the elucidation of the mechanism of onset. These factors, which include electrophysiological and biochemical markers, are also expected to facilitate the early diagnosis of diseases, prediction of prognosis, and development of treatments.

Regarding clinical symptoms and disease course, the interval from symptom onset to the first evaluation was a significant factor influencing prognosis. One recent study reported that the disease duration at entry is one of the variables that can be used to discriminate slow, average, and fast progressors [47]. Among the onset symptoms, although bulbar onset has been reported to be related to shorter survival [48]. it is not an independent predictor of outcome by multivariable analyses [49]. We found that the onset symptoms did not differ significantly between patients with fast and slow progression.

Our study had several limitations. We did not examine the firing frequency of the FPs in each muscle. Differences in the firing frequency of FPs between ALS and benign fasciculations are valuable [12], but diagnosis may be difficult when using the firing frequency of FPs. However, because the firing frequency of FPs has been reported to correlate with disease progression, evaluation of FPs in a fixed period may be required. Furthermore, the examined muscles were determined according to the revised El Escorial criteria and Awaji criteria, however, because this was a retrospective study, the examined muscles and numbers differed among the patients. In our study, we evaluated the prognosis of patients using the severity of ALS as an index, which indicated whether they would die or develop respiratory failure. This study was also retrospective. Therefore, the standard ALSFRS-R, which is a clinically useful functional assessment scale, was not suitable for this study. In addition, it was not possible to incorporate evaluation items based on the presence or absence of muscle weakness. Further, the sample was small, and the findings of the study may not be generalizable. The retrospective nature of the study may have also introduced bias. While it is important to include data from the healthy control group for comparative analysis, we were unable to analyze the relationship between FPs and uric acid levels by adding data from the healthy control group. FPs can be observed in healthy individuals (benign fasciculation), but they are distinguished by the absence of accompanying muscle weakness and the simple waveform [14, 21]. When we performed needle electromyography, FPs were rarely observed in healthy individuals. Additionally, it has been reported that uric acid concentrations are significantly lower in patients with ALS than in individuals without it [37], but they varied within the normal range in this study.

## Conclusions

The electrophysiological findings indicated degeneration from the peripheral to the central nervous system in ALS, and they were different among the groups classified by prognosis. Although this was a small study, the subjects were classified according to their real survival durations. This study revealed that the percentage of muscles with FPs and UA levels are reliable and useful prognostic factors. Particularly, high percentage of FPs observed on EMG is related to rapid progression of ALS. As described in previous studies, UA levels are negatively correlated with ALS prognosis. Surprisingly, our results demonstrated that the percentage of muscles with FPs had a significant negative relationship with UA levels. Collectively, our findings indicate that electrophysiological tests with needle EMG and laboratory data can predict the prognosis of ALS precisely, and will be beneficial for guiding therapeutic strategies for ALS.

## Supporting information

**S1 Fig. Comparison of the survival curves after disease onset stratified by rate of muscle with fibrillation potentials and positive sharp waves (Fib/PSWs) in ALS patients by using Kaplan–Meier method.**
(DOCX)

**S2 Fig. The survival curves for ALS patients with albumin $<$ 4 g/dL vs. albumin $>$ 4.1 g/dL using Kaplan–Meier method.**
(DOCX)

**S3 Fig.** The survival curves for male ALS patients with total cholesterol $<$ 189 mg/dL vs. total cholesterol $>$ 190 mg/dL using Kaplan–Meier method (A). The survival curves for female ALS patients with total cholesterol $<$ 199 mg/dL vs. total cholesterol $>$ 200 mg/dL using Kaplan–Meier method (B).
(DOCX)

**S4 Fig.** The survival curves for male ALS patients with creatine kinase $<$ 180 U/L vs. creatine kinase $>$ 181 U/L using Kaplan–Meier method (A). The survival curves for female ALS patients with creatine kinase $<$ 80 U/L vs. creatine kinase $>$ 81 U/L using Kaplan–Meier method (B).
(DOCX)

**S5 Fig. The survival curves for ALS patients with triglyceride $<$ 90 mg/dL vs. triglyceride $>$ 91 mg/dL using Kaplan–Meier method.**
(DOCX)

**S6 Fig.** The survival curves for male ALS patients with creatinine $<$ 0.6 mg/dL vs. creatinine $>$ 0.61 mg/dL by using Kaplan–Meier method (A). The survival curves for female ALS patients with creatinine $<$ 0.5 mg/dL vs. creatinine $>$ 0.51 mg/dL by using Kaplan–Meier method (B).
(DOCX)

**S7 Fig. The survival curves for ALS patients with neutrophils $<$ 3499/μL vs. neutrophils $>$ 3500/μL using Kaplan–Meier method.**
(DOCX)

## Acknowledgments

The authors express their appreciation to the laboratory technicians in the study center for their technical support in performing electrophysiological studies.

## Author Contributions

**Conceptualization:** Keiko Ohnari, Hiroaki Adachi.

**Data curation:** Keiko Ohnari, Hiroaki Adachi.

**Formal analysis:** Keiko Ohnari, Kosuke Mafune, Hiroaki Adachi.

**Investigation:** Keiko Ohnari, Hiroaki Adachi.

**Methodology:** Keiko Ohnari, Kosuke Mafune, Hiroaki Adachi.

**Resources:** Keiko Ohnari.

**Supervision:** Hiroaki Adachi.

**Validation:** Keiko Ohnari, Kosuke Mafune.

**Writing – original draft:** Keiko Ohnari, Hiroaki Adachi.

**Writing – review & editing:** Keiko Ohnari, Hiroaki Adachi.

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
