## [Decision Letter · Decision Letter 0]

19 Jun 2024

PONE-D-24-18589Fasciculation potentials are related to the prognosis of amyotrophic lateral sclerosisPLOS ONE

Dear Dr. Adachi,

Thank you for submitting your manuscript to PLOS ONE. After careful consideration, we feel that it has merit but does not fully meet PLOS ONE’s publication criteria as it currently stands. Therefore, we invite you to submit a revised version of the manuscript that addresses the points raised during the review process.

We look forward to receiving your revised manuscript.

Kind regards,

Aditya Kumar Padhi, Ph.D.

Academic Editor

PLOS ONE

Reviewers' comments:

Reviewer's Responses to Questions

**Comments to the Author**

1. Is the manuscript technically sound, and do the data support the conclusions?

Reviewer #1: Yes

Reviewer #2: Partly

Reviewer #3: No

2. Has the statistical analysis been performed appropriately and rigorously? 

Reviewer #1: Yes

Reviewer #2: Yes

Reviewer #3: No

3. Have the authors made all data underlying the findings in their manuscript fully available?

Reviewer #1: Yes

Reviewer #2: Yes

Reviewer #3: Yes

4. Is the manuscript presented in an intelligible fashion and written in standard English?

Reviewer #1: Yes

Reviewer #2: Yes

Reviewer #3: No

5. Review Comments to the Author

Reviewer #1: This article offers insights into the prognostic factors of amyotrophic lateral sclerosis (ALS). The study discovered that a higher percentage of muscles exhibiting fasciculation potentials (FPs) during EMG examinations, combined with lower uric acid (UA) levels, are strongly linked to faster disease progression and reduced survival times in ALS patients. These results underscore the potential of utilizing needle EMG and UA levels as dependable and practical biomarkers for forecasting ALS outcomes. While the findings are interesting, several issues need to be addressed for clarity and completeness.

1. Classification: The authors should better justify their classification into faster, medium and slower progressors. How did they set the time limit?

2. EMG: it seems that the authors did not follow a specific protocol with definite muscles to explore. Additionally, it is crucial to know whether the muscles examined were clinically affected or subclinically involved. As a result, the percentage of muscles exhibiting fasciculation or fibrillation potentials becomes a relative measure, undermining the reliability of the findings. This inconsistency needs to be addressed to ensure comparability and accuracy of the results across all subjects. Any association between the morphology of FPs and prognosis/survival?

3. I have concerns about the use of the trapezius muscle instead of the genioglossus muscle for EMG. Although the genioglossus is difficult for patients to relax, it is more representative of bulbar muscles, which are crucial in ALS assessment. In contrast, the trapezius muscle is more associated with the upper cervical roots rather than bulbar regions. This choice may not accurately reflect bulbar involvement and could affect the study's findings and conclusions.

4. UA: the authors should better describe the biological reason of using UA, and not another routine biomarker, in this study.

5. Sensory conduction and symptoms: the authors did not mention a very recent study describing that sensory symptoms and SNAP amplitude parallel motor disability in ALS patients, trough the classification by King’s stage. I suggest the author to perform a similar analysis, stratifying patients according to the clinical stages. Again, the sentence: “Although patients with ALS do not exhibit sensory disturbances” I don’t think is correct. Please rephrase. Pathological results on sensory dysfunction are based non only on sural nerve biopsy but also on skin biopsy as recently demonstrated, please update the literature on the topic.

Minor:

Typos: There is an error in the naming of supplementary materials.

Introduction: please note that among poor prognostic factors, the authors should also include autonomic dysfunction as recently described. Please add this info.

Reviewer #2: PONE-D-24-18589: Fasciculation potentials are related to the prognosis of amyotrophic lateral sclerosis

The manuscript presents a study investigating the relationship between the proportion of muscles with fasciculation potentials (FPs), various biochemical markers, and the prognosis of ALS. The findings are significant and well-supported by the data, demonstrating clear associations between the percentage of muscles with FPs, uric acid (UA) levels, and ALS progression.With the suggested major revisions and clarifications, the manuscript will be well-positioned to make a substantial contribution to the field of ALS research.

Below are my comments:

Major comments

1. The manuscript is missing one of the most important progressions measuring tool, ALS Functional Rating Scale-Revised (ALSFRS-R). This tool is widely used clinical tool that assesses functional abilities across four domains: bulbar function, fine motor function, gross motor function, and respiratory function. The ALS FR should be provided for all patients and the disease progression and the findings in this manuscript should be clearly correlated. Include data on functional outcomes using ALSFRS-R scores to correlate the findings with patient quality of life and functional decline. This will help validate the clinical significance of FPs and UA levels.

2. Assess the firing frequency and other characteristics of FPs. This could differentiate between benign fasciculations and those indicative of ALS progression. Such detailed analysis can add depth to the understanding of FPs in ALS.

3. Is the genetic data available for these patients? Most importantly the commonly associated mutations with ALS (e.g., SOD1, C9orf72) to see if there is any correlation with FPs and UA levels. This could provide insights into the underlying mechanisms.

4. The control data is missing. Include a healthy control group for comparison to clearly delineate the differences in FPs and UA levels between ALS patients and healthy individuals. This will strengthen the argument for the specificity of these markers in ALS. Comparisons should be made between healthy controls and patients for making any associations between findings.

5. The discussion needs more details of the potential mechanisms underlying the associations found.

Minor comments

1. Consider providing a brief overview of ALS, its pathophysiology, and the significance of identifying prognostic markers.

2. The rationale for focusing on FPs and UA levels is well-articulated, but it might benefit from a brief explanation of how these markers have been studied in previous studies.

3. Clearly classify the patients are possible, probable and definite ALS.

4. Mention the specific criteria used for defining FPs in more detail, including any relevant thresholds or cut-off values.

5. Address the study's limitations more comprehensively, including potential biases and generalizability issues.

6. Suggest directions for future research, particularly regarding the clinical application of your findings.

Reviewer #3: 1. Time from symptom onset to the first evaluation significantly differed between the groups. However, the difference was not taken into account in the analysis shown in Table 5.

2. The authors reported the correlation between the FP rate and the UA level. However, biological relevance of this finding is unclear. It appears to be spurious corrlation.

6. PLOS authors have the option to publish the peer review history of their article (what does this mean?). If published, this will include your full peer review and any attached files.

Reviewer #1: No

Reviewer #2: No

Reviewer #3: No

---

## [Author Response · Author response to Decision Letter 0]

3 Sep 2024

General comments to the Reviewers:

We thank the three expert Reviewers for their valuable feedback on our manuscript and sincerely appreciate their time and efforts. Their comments have been invaluable. In response to the Reviewer’s comments and suggestions, we have thoroughly revised our manuscript. For clarity, the revised portions of the manuscript are indicated in red. We've addressed each of your specific comments separately. We appreciate your continued consideration and look forward to any further insights.

Below, we provide a point-by-point response to each of the Reviewers’ comments:

Reviewer #1: This article offers insights into the prognostic factors of amyotrophic lateral sclerosis (ALS). The study discovered that a higher percentage of muscles exhibiting fasciculation potentials (FPs) during EMG examinations, combined with lower uric acid (UA) levels, are strongly linked to faster disease progression and reduced survival times in ALS patients. These results underscore the potential of utilizing needle EMG and UA levels as dependable and practical biomarkers for forecasting ALS outcomes. While the findings are interesting, several issues need to be addressed for clarity and completeness.

1. Classification: The authors should better justify their classification into faster, medium and slower progressors. How did they set the time limit?

Response: We thank the Reviewer for this constructive feedback on classification. The median survival duration of the patients with ALS was 3 years, and the 1-year survival rate was 76.2% (Reference 2). Therefore, we considered 3 years as the average survival duration and 1 year as short duration associated with rapid progression of the disease. We classified the survival durations into 3: less than 1 year: fast progression; from 1 year to less than 3 years: average progression; and 3 years or more: slow progression. We have added the following sentences to the first paragraph of the Methods section accordingly (lines 86–87): "The median survival duration of the patients with ALS was 3 years, and the 1-year survival rate was 76.2%."

2. EMG: it seems that the authors did not follow a specific protocol with definite muscles to explore. Additionally, it is crucial to know whether the muscles examined were clinically affected or subclinically involved. As a result, the percentage of muscles exhibiting fasciculation or fibrillation potentials becomes a relative measure, undermining the reliability of the findings. This inconsistency needs to be addressed to ensure comparability and accuracy of the results across all subjects. Any association between the morphology of FPs and prognosis/survival?

Response: We thank the Reviewer for this constructive criticism of our manuscript. We selected muscles from four body regions according to the diagnostic criteria for ALS. However, this study was retrospective, as you pointed out. The problem was that different muscles were used for different cases, and the influence of the presence or absence of muscle weakness was not considered. We have added the following sentences to the limitations section of the study (lines 354-356): " In addition, it was not possible to incorporate evaluation items based on the presence or absence of muscle weakness.” Further, since it has been reported that the morphology of FP is related to prognosis, we evaluated polyphasic FP. The methods are described (lines 118–121).

3. I have concerns about the use of the trapezius muscle instead of the genioglossus muscle for EMG. Although the genioglossus is difficult for patients to relax, it is more representative of bulbar muscles, which are crucial in ALS assessment. In contrast, the trapezius muscle is more associated with the upper cervical roots rather than bulbar regions. This choice may not accurately reflect bulbar involvement and could affect the study's findings and conclusions.

Response: We are grateful to the Reviewer for the valuable comments. As you pointed out, the trapezius muscle is also controlled by the upper cervical roots, so it may not be appropriate to evaluate it as a cranial nerve muscle. However, on the other hand, active denervation findings for the trapezius muscle are highly specific to ALS. This study focused on the trapezius muscle because needle electromyography is easy to perform on it and the burden on patients is light. We have included text regarding the selection of the muscles to be tested in the methods section (lines 112–115).

4. UA: the authors should better describe the biological reason of using UA, and not another routine biomarker, in this study.

Response: This point is one that the Reviewer would naturally have doubts about. As pointed out by the Reviewer, we also investigated the effects of albumin, triglycerides, total cholesterol, creatinine, neutrophil count, and creatine kinase, among others. These have been reported to be related to the prognosis of ALS. We also investigated the relationship between these markers and prognosis. The results are shown in Figures S2–S7. However, only uric acid demonstrated an association with prognosis, and we further investigated it.

5. Sensory conduction and symptoms: the authors did not mention a very recent study describing that sensory symptoms and SNAP amplitude parallel motor disability in ALS patients, trough the classification by King’s stage. I suggest the author to perform a similar analysis, stratifying patients according to the clinical stages. Again, the sentence: “Although patients with ALS do not exhibit sensory disturbances” I don’t think is correct. Please rephrase. Pathological results on sensory dysfunction are based non only on sural nerve biopsy but also on skin biopsy as recently demonstrated, please update the literature on the topic.

Response: We would like to thank the Reviewer for these constructive comments. As pointed out, there have been important reports on the relationship between the stage of ALS and SNAP amplitude recently. Therefore, the sentence “Although patients with ALS do not exhibit sensory disturbances” has been corrected to “Although patients with ALS do not usually show sensory disturbances” (line 290). In addition, sensory neuropathy in ALS has been pathologically proven using skin biopsy tissue. We have added the following sentences to the discussion section (lines 298-300): "Pathological sensory disturbance detected through skin biopsies has also been reported for ALS, with observations including axonal swellings as the foremost morphological changes and small fiber regeneration insufficiency."

Minor:

Typos: There is an error in the naming of supplementary materials.

Response: We thank the Reviewer for pointing this out. We have corrected the error in the naming of supplementary materials.

Introduction: please note that among poor prognostic factors, the authors should also include autonomic dysfunction as recently described. Please add this info.

Response: We thank the Reviewer for this suggestion. We have added autonomic dysfunction as a prognostic factor for ALS in the Introduction (line 55).

Reviewer #2: PONE-D-24-18589: Fasciculation potentials are related to the prognosis of amyotrophic lateral sclerosis

The manuscript presents a study investigating the relationship between the proportion of muscles with fasciculation potentials (FPs), various biochemical markers, and the prognosis of ALS. The findings are significant and well-supported by the data, demonstrating clear associations between the percentage of muscles with FPs, uric acid (UA) levels, and ALS progression. With the suggested major revisions and clarifications, the manuscript will be well-positioned to make a substantial contribution to the field of ALS research.

Below are my comments:

Major comments

1. The manuscript is missing one of the most important progressions measuring tool, ALS Functional Rating Scale-Revised (ALSFRS-R). This tool is widely used clinical tool that assesses functional abilities across four domains: bulbar function, fine motor function, gross motor function, and respiratory function. The ALS FR should be provided for all patients and the disease progression and the findings in this manuscript should be clearly correlated. Include data on functional outcomes using ALSFRS-R scores to correlate the findings with patient quality of life and functional decline. This will help validate the clinical significance of FPs and UA level

Response: We thank the Reviewer for the insightful comments and suggestions. As pointed out, the ALSFRS-R is a clinically useful functional evaluation scale and is considered an essential scale in clinical trials for evaluating the therapeutic effects of drugs and the differences in functional prognosis due to related factors. On the other hand, in our current research, we evaluated the prognosis of ALS using the severity of the disease, which indicated whether a patient would die or develop respiratory failure. Therefore, it is considered slightly different from the adaptation of ALSFRS-R, which can sensitively detect gradual changes in function. Additionally, this study was retrospective, and it would be difficult to evaluate ALSFRS-R retrospectively. Therefore, we have added the following sentences related to the adaptation of ALSFRS-R to the research limitations (lines 351-354): “In our study, we evaluated the prognosis of patients using the severity of ALS as an index, which indicated whether they would die or develop respiratory failure. This study was also retrospective. Therefore, the standard ALSFRS-R, which is a clinically useful functional assessment scale, was not suitable for this study.”

2. Assess the firing frequency and other characteristics of FPs. This could differentiate between benign fasciculations and those indicative of ALS progression. Such detailed analysis can add depth to the understanding of FPs in ALS.

Response: We thank the Reviewer for the insightful comments and valuable suggestions. As pointed out, the firing frequency and morphology of FPs were related to the progression of ALS. However, it is very difficult to accurately measure the firing frequency of FPs in muscles of a certain size, and we do not usually measure the firing frequency in clinical practice. However, regarding morphology, electromyography is performed to exclude benign fasciculations and to not count FPs that show only a single phase. These points are discussed in the Methods and Limitations section (lines 119–121 and 344-348).

3. Is the genetic data available for these patients? Most importantly the commonly associated mutations with ALS (e.g., SOD1, C9orf72) to see if there is any correlation with FPs and UA levels. This could provide insights into the underlying mechanisms.

Response: We thank the Reviewer for the important comments. Examining whether mutations commonly associated with ALS (e.g., SOD1, C9orf72) influence the correlation between FPs and UA levels and prognosis will yield clinically important data about the underlying mechanisms. Unfortunately, we targeted patients who were thought to have sporadic ALS and were negative for commonly associated mutations in our study, and patients with a family history were excluded. We have added that we targeted patients with sporadic ALS (line 85).

4. The control data is missing. Include a healthy control group for comparison to clearly delineate the differences in FPs and UA levels between ALS patients and healthy individuals. This will strengthen the argument for the specificity of these markers in ALS. Comparisons should be made between healthy controls and patients for making any associations between findings.

Response: We thank the Reviewer for this constructive criticism of our manuscript. As pointed out, we also believe that analysis that includes comparison with a healthy control group is important. Although FPs are observed in normal people (benign fasciculation), they are distinguished by the absence of muscle weakness and simple waveforms (References 14,21). FPs are extremely rare in healthy people when we perform needle electromyography, and we were unable to provide data for the healthy control group in this study. On the other hand, it has been reported that uric acid levels are significantly lower for the ALS group than for the control group (Reference 37). In our study, the ALS group, which had rapid progression, showed lower values. However, the fluctuations in uric acid levels were within the normal range for general uric acid levels. Therefore, we did not include a comparison with a healthy control group in this analysis. We have added the following sentences to the limitations section regarding the fact that we were unable to analyze the relationship between FPs and uric acid levels by adding a comparison with a healthy control group (lines 357-365): “While it is important to include data from the healthy control group for comparative analysis, we were unable to analyze the relationship between FPs and uric acid levels by adding data from the healthy control group. FPs can be observed in healthy individuals (benign fasciculation), but they are distinguished by the absence of accompanying muscle weakness and the simple waveform [14,21]. When we performed needle electromyography, FPs were rarely observed in healthy individuals. Additionally, it has been reported that uric acid levels are significantly lower in patients with ALS than in individuals without it [37], but they varied within the normal range in this study.”

5. The discussion needs more details of the potential mechanisms underlying the associations found.

Response: We thank the Reviewer for this valuable suggestion. Unfortunately, we were unable to conduct an objective discussion using the literature, as no reports on the relationship between FPs and serum UA levels in ALS have been published. However, from our data, serum UA levels are involved in the mechanism underlying electrophysiological changes that lead to FPs in skeletal muscle and may modify the pathology of ALS. We have added the following sentences to the Discussion section (lines 320-324): “No reports on the relationship between FPs and serum UA levels in ALS have been published to date. However, serum UA levels and FPs were negatively correlated in this study. This suggests that serum UA levels s may be involved in the mechanism underlying electrophysiological changes that lead to FPs in skeletal muscle and may modify the pathology of ALS.”

Minor comments

1. Consider providing a brief overview of ALS, its pathophysiology, and the significance of identifying prognostic markers.

Response: Thank you for your valuable suggestions. We have added the following sentences to the Discussion section (lines 331-336): “Sporadic ALS is an intractable disease, and its mechanism of onset is still unknown. The discovery of several factors involved in the pathogenesis or modification of the disease progression of ALS may lead to the elucidation of the mechanism of onset. These factors, which include electrophysiological and biochemical markers, are also expected to facilitate the early diagnosis of diseases, prediction of prognosis, and development of treatments.”

2. The rationale for focusing on FPs and UA levels is well-articulated, but it might benefit from a brief explanation of how these markers have been studied in previous studies.

Response: Unfortunately, we were unable to conduct an objective discussion using the literature, as no reports on the relationship between FPs and serum UA concentrations in ALS have been published.

3. Clearly classify the patients are possible, probable and definite ALS.

Response: This study included patients who could be followed up to death or respiratory support, and they were classified as having definite ALS. We have added this information (line 85).

4. Mention the specific criteria used for defining FPs in more detail, including any relevant thresholds or cut-off values.

Response: As pointed out, we have added that FPs in needle electromyography are identified by polyphasic waves that appear spontaneously and randomly (lines 119-121). 

5. Address the study's limitations more comprehensively, including potential biases and generalizability iss

---

## [Decision Letter · Decision Letter 1]

22 Sep 2024

PONE-D-24-18589R1Fasciculation potentials are related to the prognosis of amyotrophic lateral sclerosisPLOS ONE

Dear Dr. Adachi,

Thank you for submitting your manuscript to PLOS ONE. After careful consideration, we feel that it has merit but does not fully meet PLOS ONE’s publication criteria as it currently stands. Therefore, we invite you to submit a revised version of the manuscript that addresses the points raised during the review process.

We look forward to receiving your revised manuscript.

Kind regards,

Aditya Kumar Padhi, Ph.D.

Academic Editor

PLOS ONE

Journal Requirements:

Reviewers' comments:

Reviewer's Responses to Questions

**Comments to the Author**

1. If the authors have adequately addressed your comments raised in a previous round of review and you feel that this manuscript is now acceptable for publication, you may indicate that here to bypass the “Comments to the Author” section, enter your conflict of interest statement in the “Confidential to Editor” section, and submit your "Accept" recommendation.

Reviewer #1: All comments have been addressed

Reviewer #2: All comments have been addressed

Reviewer #3: All comments have been addressed

2. Is the manuscript technically sound, and do the data support the conclusions?

Reviewer #1: Yes

Reviewer #2: Partly

Reviewer #3: Yes

3. Has the statistical analysis been performed appropriately and rigorously? 

Reviewer #1: Yes

Reviewer #2: Yes

Reviewer #3: Yes

4. Have the authors made all data underlying the findings in their manuscript fully available?

Reviewer #1: Yes

Reviewer #2: Yes

Reviewer #3: Yes

5. Is the manuscript presented in an intelligible fashion and written in standard English?

Reviewer #1: Yes

Reviewer #2: Yes

Reviewer #3: Yes

6. Review Comments to the Author

Reviewer #1: The authors have done a commendable job with the revisions, but I would like to point out a minor issue in the following sentence: 'Pathological sensory disturbance detected through skin biopsies has also been reported for ALS, with observations including axonal swellings as the foremost morphological changes and small fiber regeneration insufficiency.' The authors cite an older study that does not clearly highlight the extent of cutaneous denervation or the progression of SNAP amplitudes across different disease stages. I would kindly request that the authors revise this sentence, citing a more appropriate and recent study about skin innervation across amyotrophic lateral sclerosis clinical stages.

Reviewer #2: All my concerns and comments have been answered. The manuscript seems to have considerably improved after inclusion of author response.

Reviewer #3: The review comments at the previous round have been addressed. The reviewer has no additional comments.

7. PLOS authors have the option to publish the peer review history of their article (what does this mean?). If published, this will include your full peer review and any attached files.

Reviewer #1: No

Reviewer #2: No

Reviewer #3: No

---

## [Author Response · Author response to Decision Letter 1]

6 Oct 2024

Reviewer #1: The authors have done a commendable job with the revisions, but I would like to point out a minor issue in the following sentence: 'Pathological sensory disturbance detected through skin biopsies has also been reported for ALS, with observations including axonal swellings as the foremost morphological changes and small fiber regeneration insufficiency.' The authors cite an older study that does not clearly highlight the extent of cutaneous denervation or the progression of SNAP amplitudes across different disease stages. I would kindly request that the authors revise this sentence, citing a more appropriate and recent study about skin innervation across amyotrophic lateral sclerosis clinical stages.

Response: We thank the Reviewer for this constructive feedback on pathological sensory disturbance detected through skin biopsies in ALS. As you pointed out, we have cited other, more recent study and revised the following sentence in the fifth paragraph of the discussion part: “Pathological sensory disturbance detected through skin biopsies has also been reported for ALS, with observations including, a significant loss of intraepidermal nerve fiber (IFNF) and Meissner corpuscle density in ALS compared with healthy controls, and increasing IENF density over time were associated with a poorer prognosis” (line 298-301).

---

## [Decision Letter · Decision Letter 2]

23 Oct 2024

Fasciculation potentials are related to the prognosis of amyotrophic lateral sclerosis

PONE-D-24-18589R2

Dear Dr. Adachi,

We’re pleased to inform you that your manuscript has been judged scientifically suitable for publication and will be formally accepted for publication once it meets all outstanding technical requirements.

Kind regards,

Aditya Kumar Padhi, Ph.D.

Academic Editor

PLOS ONE

Additional Editor Comments (optional):

Reviewers' comments:

Reviewer's Responses to Questions

**Comments to the Author**

1. If the authors have adequately addressed your comments raised in a previous round of review and you feel that this manuscript is now acceptable for publication, you may indicate that here to bypass the “Comments to the Author” section, enter your conflict of interest statement in the “Confidential to Editor” section, and submit your "Accept" recommendation.

Reviewer #1: All comments have been addressed

2. Is the manuscript technically sound, and do the data support the conclusions?

Reviewer #1: Yes

3. Has the statistical analysis been performed appropriately and rigorously? 

Reviewer #1: Yes

4. Have the authors made all data underlying the findings in their manuscript fully available?

Reviewer #1: Yes

5. Is the manuscript presented in an intelligible fashion and written in standard English?

Reviewer #1: Yes

6. Review Comments to the Author

Reviewer #1: The authors have adequately addressed all of my previous comments and concerns. I have no further requests or suggestions to submit.

7. PLOS authors have the option to publish the peer review history of their article (what does this mean?). If published, this will include your full peer review and any attached files.

Reviewer #1: No

---

## [Editor Report · Acceptance letter]

30 Oct 2024

PONE-D-24-18589R2 

PLOS ONE

Dear Dr. Adachi, 

I'm pleased to inform you that your manuscript has been deemed suitable for publication in PLOS ONE. Congratulations! Your manuscript is now being handed over to our production team.

Kind regards, 

on behalf of

Dr. Aditya Kumar Padhi 

Academic Editor

PLOS ONE